# Welfare and Self-Assessment in Patients after Aesthetic and Reconstructive Treatments

**DOI:** 10.3390/ijerph191811238

**Published:** 2022-09-07

**Authors:** Edyta Skwirczyńska, Michał Piotrowiak, Marek Ostrowski, Oskar Wróblewski, Karol Tejchman, Sebastian Kwiatkowski, Aneta Cymbaluk-Płoska

**Affiliations:** 1Department of History of Medicine and Ethics, Pomeranian Medical University, Rybacka 1, 70-204 Szczecin, Poland; 2Department of General and Transplantation Surgery, Pomeranian Medical University, al. Powstańców Wielkopolskich 72, 70-111 Szczecin, Poland; 3Department of Obstetrics and Gynecology, Pomeranian Medical University, al. Powstańców Wielkopolskich 72, 70-111 Szczecin, Poland; 4Department of Gynecological Surgery and Gynecological Oncology of Adults and Adolescents, Pomeranian Medical University, al. Powstańców Wielkopolskich 72, 70-111 Szczecin, Poland

**Keywords:** aesthetic surgery, self-assessment, Body Esteem Scale, Rosenberg Self-Esteem Scale, Satisfaction with Life Scale

## Abstract

In the last decade, there has been a noticeable increase in the interest in aesthetic and corrective surgery regardless of a patient’s age. Both aesthetical and practical considerations are a motivation for patients undergoing plastic surgery. The goal of this study is to analyze dependencies between welfare, self-assessment and body self-perception in patients that qualified for plastic and aesthetic surgical procedures. The study group included 164 female patients, of whom 124 patients filled out a questionnaire before and after surgery. The questionnaire included demographic data and scales such as the Body Esteem Scale, the Rosenberg Self-Esteem Scale—SES, the Satisfaction with Life Scale—SWLS, the Flourishing Scale and the Scale of Positive and Negative Experience—SPANE. The first hypothesis concerned the subjective assessment of body self-perception after the procedure. The results of the study confirm this hypothesis—female patients after surgery rate their body self-perception higher, which indicates a positive influence of plastic and aesthetic surgery that increased in the subjective assessment of 66 examined patients. Moreover, the study revealed a higher self-assessment after procedures. On the other hand, the results indicated that younger patients had a higher body assessment, but there was no increase in self-assessment. Except for breast augmentation surgery, there was no influence on self-assessment and life satisfaction improvement after other surgical procedures. In patients up to 48 years old, after surgery, there was a significant dependence between subjective body self-assessment and all surveyed forms of welfare. In the case of patients after 48 year of age, there was a relationship between life satisfaction and body self-perception both before and after surgical treatment.

## 1. Introduction

Since the dawn of time, man has followed what is beautiful and perfect. The approach to beauty represented by representatives of various epochs has gradually changed; however, it is not difficult to notice some universal sets of features that make up the global assessment of an object or person. Umberto Eco in his book, “A History of Beauty”, indicates the coincidence of a set of features that coexist with the word “beautiful”, enumerating such expressions as gratitude, sublime, order and wonder [1]. One can speak of a pattern taken when defining what is beautiful, but the assessment result will change depending on the geographical area, environment, and cultural patterns [2]. Attention should be paid to the subjective nature of the assessments issued depending on the subject, the perspective changes, and the final result of the assessment with it. The first mentions of correcting beauty are from Egyptian papyri that date back to the 16th century BC. It is evident that bringing the face back to its original state after it was damaged as a result of aging processes or mechanical damage was present in human beings almost from the beginning of its existence [3]. In the last decade, there has been a noticeable increase in interest in corrective and aesthetic plastic surgery, regardless of the patient’s age. The reasons for undergoing the treatments are different and may be related to aesthetic and practical reasons. The main factors responsible for the popularization of cosmetic surgery among patients are dynamic technological developments, improvements in the safety of the procedure and shortening the convalescence time. Another factor influencing a significant increase in patients’ interest is popularization in the mass media [4]. Social transformations make beauty one of the most important factors responsible for success. The constantly changing canons of beauty mean that people who want to become part of them have to use the help of appropriate specialists. The self-esteem of these people is not without significance, as it was the increased interest in plastic surgery that caused researchers dealing with aspects related to the psychosocial functioning of a person to start to analyze the characteristics of patients deciding to make corrections to their own body [5].

Patients deciding to undergo a procedure in the field of aesthetic medicine are most often guided by criteria related to the functional improvement of the body parts that cause them discomfort, whereas aesthetic reasons are another motivation. Regardless of their motivation, patients treat the surgeon as a vehicle of hope for improving not only their image, but also their own self-esteem and satisfaction with life [6]. In recent years, there has been a noticeable trend of choosing plastic surgeries without health indications to perform them. In healthy patients, the decision to undergo the procedure is most often dictated by the improvement in both the physical and mental well-being, and the aspect related to social acceptance is not without significance. The American Medical Association has defined aesthetic medicine treatments as the transformation of normal body structures in order to improve the patient’s appearance and self-esteem [7]. The developed definition emphasizes not only the successful transformation of body structures in accordance with the patient’s requirements, but above all emphasizes the improvement of his mental well-being as one of the important post-treatment effects. It seems that the surgeon performing the procedure should not ignore its psychological implications. On the other hand, when already at the stage of planning the procedure itself, it is necessary to consider the psychological context in order to achieve patient satisfaction.

The aforementioned psychological issues related to the procedure predispose a pre-procedure consultation as a critical factor affecting the patient’s later satisfaction. The psychological profile of patients who decide to undergo plastic surgery has significantly changed over time. The first studies in the 1950s and 1960s, based on patient interviews, indicated some form of psychological pathology. Only on the basis of standardized psychometric tests was it shown that the share of psychopathological factors was lower than originally assumed. The results of the research carried out at that time demonstrated a lower percentage of psychological disorders in patients deciding to undergo plastic surgery compared to the population than was originally presented.

Patients opting for plastic surgery had more social and emotional concerns about their facial appearance than the rest of the population. Increased emphasis on one’s own appearance was one of the factors that led to the decision to undergo the procedure; at the same time, the objections related to one’s own appearance were most often devoid of pathological features. A low frequency of psychosocial disorders in the population undergoing aesthetic medicine treatments has also been indicated [6]. By analyzing some specific features, studies have shown that in the case of patients undergoing a facelift, the most numerous groups are middle-aged people in the period of a personality crisis related to children leaving the family home or divorce. When analyzing patients undergoing aesthetic surgery, it is impossible to identify one factor. As mentioned before, it is assumed that patients are divided according to both objective and subjective factors. Another proposal regarding the motives for undergoing plastic surgery is the image of the patient’s body, usually consisting of two elements. The first element concerns the orientation of the body image, understood as the significance of the body image from the patient’s point of view and the assessment of the body image understood as its global assessment by the patient [8,9].

For a more complete analysis to understand the motivation of patients to undergo plastic surgery, it is necessary to detail the factors that may individually or jointly lead to a positive decision regarding the procedure. First, we discuss women who decide to undergo breast reconstruction after mastectomy. Among this group of women, there are intrapsychic, informational, medical, interpersonal and economic factors.

Intrapsychic factors are related to the internal motivations of patients in connection with the breast augmentation procedure, the effect of which is a change in the image of the patient’s own body, which also positively affects the patient’s quality of life. Considerations concerning the intrapsychic factor cannot ignore the emotional sphere of patients in relation to their physical appearance, because both the size and shape of the breasts are main parts responsible for the identity of a woman [10]. The common denominator connecting women deciding to undergo breast surgery was dissatisfaction with a specific part of the body, which translated into general dissatisfaction with their own appearance. These results may support the hypothesis about the actual motivation to undergo the procedure, which is the result of global dissatisfaction with one’s body image caused by a specific feature. When analyzing the conducted research, it can be concluded that preoperative patients are more insecure about their breasts, comparing their appearance to other women. Patients also tend to avoid situations when those in their surroundings could see them without clothes, masking their breasts with additional garments or checking the appearance of their breasts several times a day [11].

Another group of factors is constituted by medical and informational variables that have an equally significant impact on a patient’s decision-making as intrapsychic factors. The medical factors include the patient’s health condition and her health habits, whereas the informational factors concern the state of knowledge about the procedure itself, as well as the possible complications and the degree of risk associated with the procedure [12].

Another important factor influencing the decisions of patients is the economic factor. Due to the relatively high cost of the procedure and, in most cases, the lack of funding from the National Health Fund, patients allocate their own funds, use credit institutions or make intra-family loans to finance the procedure.

The last factor classified as interpersonal is related to interpersonal relationships and the ability to create lasting relationships. Research shows that patients considering breast surgery have a poorer emotional life, less frequently use contraceptives and have a smaller number of sexual partners [1,13].

Individual categories of factors in the case of breast reconstruction surgery may turn out to be similar to other types of surgery. At the moment, there are no similar studies on the motivation for a specific type of surgery; in the future, it is worth considering the gap in this area and deepening the search for motivation also in relation to other treatments.

For the purposes of this study, it can be assumed that some of the patients’ motivations may be the same in relation to patients undergoing breast reconstruction. Here, we are talking primarily about intrapsychic factors understood to improve the image of one’s own body image and material factors.

Patients’ self-esteem should also be included in the group of factors indicated above. At the beginning, it is necessary to distinguish self-esteem that can act as a motivator as well as appear in the form of a dependent variable transformed under the influence of plastic surgery [14]. Self-esteem as a psychological construct is measured using various standardized tools. The most frequently used are the Rosenberg Scale and the Multidimensional Self-Assessment Scale. Research shows a strong correlation between self-esteem and body image. As an additional element, apart from self-assessment, that could contribute to making decisions about the procedure, people from one’s social circle were indicated who could be the originators, and thus, contribute to one’s self-esteem.

Childhood complexes are another factor that may contribute to the decision to perform plastic surgery. It has been shown that people deciding to undergo surgery more often than others experienced unpleasant acts on the part of their peers. However, similar studies did not show any correlation between the decision about plastic surgery and stigmatization [15].

The last factor worth considering for the purposes of the above work is education. The research on the vision of one’s own insight indicates that 73% of people deciding to undergo surgery received a higher education, whereas 23% of the studied group received a secondary education [16].

Based on the research, there is a noticeable difference in terms of the determinants for undergoing plastic surgery. From the perspective of the doctor performing the procedure, it is important to know the patient’s motivation for the procedure. This is important because of the patient’s perceptions of the future effect. As it has already been shown, patients most often obtain information from the mass media, which may not be a good source of information. Additionally, considering the patient’s history may help the surgeon to reformulate the original assumptions of the patient in order to increase his satisfaction after the procedure. To sum up, regardless of the specific motivator, among plastic surgery patients there is a significant share of psychological factors in their treatment decisions. Considering the relatively poor literature on the motivations of patients in relation to specific types of treatments, this area may constitute a further field of exploration, contributing to the achievement of even more satisfactory post-treatment results.

There is a noticeable gap in the available literature related to the approach to studying the effects of plastic surgery procedures. The constantly growing trend related to the number of operations means that surgeons who carry out procedures are highly responsible not only for the visual effect, but also its psychological connotations. The literature on the subject proves that the decisions made by the patients do not always result from their internal beliefs. The mass media play a significant role in creating the contemporary canon of beauty, often distorting the perfect body image. Bearing this in mind, the issue of preoperative consultation is extremely important in order to compare the patient’s expectations and the possibilities offered by the surgeon with the current medical knowledge. This paper deals with the issues of psychological aspects related to aesthetic medicine treatments.

In their work, Moss and Harris showed possible methodological errors appearing in the research on the influence of plastic surgery on the psychological transformation in patients. One of the distorting factors that the authors point out is cognitive dissonance, which makes the fact of undergoing surgery induce patients to positively evaluate its results in order to justify their choice. The authors also indicate the lack of a control group [17]. Perrogon also points to the possibility of patients feeling satisfied with the procedure, whereas being dissatisfied with their own appearance, which significant correlates with variables such as quality of life, mood or behavior [18].

Self-esteem is one of the key dimensions of personality that significantly contributes to the formation of a self-image. Humans present various attitudes depending on objects, one of which is his own self. In this approach, self-esteem is a negative or positive attitude towards one’s self, which is the subject of a comprehensive self-assessment. A person’s high self-esteem may result in a belief in their own values and in being “good enough”. A person with low self-esteem feels dissatisfied with himself and rejects his own self [19]. Indicators of high self-esteem include pride and, at the same time, the ability to accept one’s own shortcomings. Positive reinforcement in the form of socially desirable behavior correlates with a sense of pride. The effect of this dependence is an increase in self-esteem and causative possibilities. People with high self-esteem tend to attribute positive traits to themselves more often. Low self-esteem is characterized by an underestimation of one’s own abilities. The perception of one’s own actions in the social context is also underestimated. People with low self-esteem experience a constant sense of shame as well as low self-esteem, threat and guilt [14]. There are many reasons for a decision to undergo surgical correction.

Patients emphasize a sense of low self-esteem and quality of life. This results from the lack of acceptance of one’s own body and emotional problems that may be the result of a discrepancy between one’s own expectations and possibilities. Patients undergoing surgery are convinced that in this way they can improve their well-being and achieve an improvement in their mental and emotional state. In a study by Darisi et al., it was found that, at the same time, they indicate the autonomy of their decisions and the awareness of the potential effect of the surgical procedure [20]. The growing interest in plastic surgeries explains their positive impact not only on the external appearance, but also the strengthening of psychological parameters such as self-esteem and well-being in life. Research shows that people after plastic surgery are more satisfied with their appearance than before surgery. Moreover, an increase in the quality of life is noticeable among these people. There is also a noticeable parameter improvement in people suffering from depression and social phobias. It should also be considered important that there were no negative psychological effects in the assessed postoperative areas among a group subjected to surgery [21]. Research shows that in patients with problems in social functioning caused by a physical defect, the results in the field of mental well-being improve after surgery. After surgery, patients return to a level of well-being comparable to the general population [22].

The aim of this study was to analyze the data obtained from patients undergoing plastic and aesthetic surgeries and the relationship between well-being, self-esteem and body evaluation.

The researchers asked the following research questions:Does the procedure in the field of plastic surgery have a positive effect on the self-esteem with regard to the patient’s body?Does the performance of the surgery affect the self-esteem of the patients?Does the age of the patients differentiate the assessment of their own body and self-esteem?Does the type of surgery performed differentiate self-esteem and life satisfaction?Does the division into age groups correlate into different forms of well-being and body assessments of patients?

## 2. Materials and Methods

In the study, 290 questionnaires from women who reported for the procedure were collected. Before the procedure, the study group consisted of 164 women (M = 45.00, SD = 14.77), whereas 126 women (M = 45.79, SD = 14.64) again completed the questionnaire at the follow-up visit six months after the surgery. The patients participating in the study were in good or very good health. The questionnaire was conducted anonymously in the form of a paper-and-pencil survey. Patients received a questionnaire with an instruction to enter the code, which included the first and last letters of their name and the full birthday date of their mother. They threw the completed questionnaires into a dedicated urn in the clinic. The individual code made it possible to adjust the questionnaires before and after the procedure while maintaining the patients’ anonymity.

The study used standardized tools and its own record questionnaire. It provided information on the patients’ age, place of residence, education and marital status. It also contained information about the type of surgery, its date and a self-report question on the history of previous treatments.

Another questionnaire was the Satisfaction with Life Scale (SWLS). The scale responds to the cognitive process of evaluating life experiences. Three components contribute to subjective well-being: positive affect, negative affect, and life satisfaction. The satisfaction assessment results were obtained by comparing the circumstances with what is considered by the person to be the appropriate standard. By means of a measurement, an overall level of satisfaction with life is obtained. It consists of five statements and a seven-point scale of answers:7—I strongly agree;6—I agree;5—I tend to agree;4—I neither agree nor disagree;3—I rather disagree;2—I disagree;1—I strongly do not agree.

The reliability of the scale is 0.82 [23].

The Flourishing Scale. This scale was created to measure eudaimonistic well-being. The scale is a development of humanistic theories relating to positive functioning. The Flourishing Scale consists of eight items with a seven-point scale of answers:7—I strongly agree;6—I agree;5—I rather agree;4—difficult to say, I neither agree nor agree;3—I rather disagree;2—I disagree;1—I strongly disagree.

The range of possible points range from 8, the lowest level of well-being, to 56, the highest level of well-being [24].

The Scale of Positive and Negative Emotions (SPANE). The scale is a 12-subject questionnaire. It consists of 6 features relating to positive feelings, 6 features describing negative feelings, and a five-point scale of answers:1—very rarely or never;2—rarely;3—sometimes;4—often;5—very often or always.

The SPANE contains a broad description of positive and negative feelings and applies to both positive and negative emotions. The overall result of the SPANE allows us to estimate the level of positive emotional balance, which is a measure of affective hedonistic well-being [24].

The SES Self-Assessment Scale was also used. The M. Rosenberg Scale, which is the Polish adaptation (Łaguna, Lachowicz-Tabaczek, Dzwonkowska), is a 10-level questionnaire used to measure global self-esteem in both adolescents and adults. The questionnaire consists of 10 diagnostic statements and a four-point scale of answers:1—I strongly agree;2—I agree;3—I disagree;4—I strongly disagree.

The overall score ranges from 10 to 40 points. The reliability of the scale depending on the age group is 0.81 to 0.83 [25].

The Polish adaptation of the Body Esteem Scale is a 35-item questionnaire used to determine the attitude of the respondents to their body. The questionnaire has three subscales relating to sexual attractiveness, weight control and physical condition. The Likert scale was used in the questionnaire, with these possible answers:1—I have strong negative feelings;2—I have moderate negative feelings;3—I have no feelings;4—I have moderate positive feelings;5—I have very positive feelings.

The reliability of the tool in the female version is 0.92 [26,27]. The STATISTICA software was used for the statistical analysis of the data.

## 3. Results

### 3.1. The Procedure

In order to investigate whether the fact of performing the procedure differentiates the level of subjective assessment of the body in the patients, an analysis was carried out using Student’s *t*-test for dependent samples. Descriptive statistics of individual subgroups for which mean differences were tested are presented in Table 1.

The analysis with Student’s *t*-test for dependent samples showed that the mean self-assessment of the patients after the procedure (M = 186.93; SD = 21.04) is statistically significantly higher than the average assessment of the patient’s own body before the procedure (M = 176, 88; SD = 22.64), as *t*(115) = −5.23; *p* < 0.001. Cohen’s d value = −0.49, which indicates the mean relationship between the patients’ subjective assessment of their own body before and after the procedure. The detailed results of the conducted analysis are presented in Table 2.

### 3.2. Self-Esteem

Through the analysis with Student’s *t*-test for dependent samples, the level of self-esteem in patients who underwent surgery was examined. Descriptive statistics of individual subgroups for which the differences of means were tested are presented in Table 3.

The analysis with Student’s *t*-test for dependent samples showed that the mean self-esteem in patients after the procedure (M = 23.61; SD = 1.98) is statistically significantly higher than the average self-esteem before the procedure (M = 22.96; SD = 2.31), as *t*(113) = −2.73; *p* < 0.05. Cohen’s d value = −0.26, which indicates a small correlation between the self-esteem of patients before and after the procedure. The detailed results of the conducted analysis are presented in Table 4.

### 3.3. Age

In order to test whether the age of the patients differentiated the level of their body assessment, an analysis was carried out using Student’s *t*-test for independent samples. In the first step of the analysis, the variable was recoded as age according to the median criterion Me = 48. As a result of the division, age variables at two levels were obtained: older patients >48 years of age, N = 149, and younger patients ≤ 48 years of age, N = 141. Descriptive statistics of individual subgroups for which mean differences were tested are presented in Table 5.

The analysis showed that the body assessment of the patients is significantly different depending on the patient’s age. Cohen’s coefficient result among younger and older patients before surgery is: *t*(162) = 4.34; *p* < 0.05; Cohen’s d = 3.09. Cohen’s coefficient for younger and older patients after surgery is: *t*(124) = 4.01; *p* < 0.05; Cohen’s d = 2.32. Among the patients before the procedure, the mean body assessment of women up to 48 years of age (M = 130.67; SD = 1.99) is 99% higher than women over 49 years of age (M = 115.37; SD = 2.96). Additionally, in the case of postoperative patients, the mean is statistically significantly differentiated depending on the age range of the patient. The average of women up to 48 years of age (M = 129.78; SD = 2.21) is 98% higher than the average of women over 49 years of age (M = 117.35; SD = 3.14). Descriptive statistics for individual groups of patients are presented in Table 5, whereas the results of the analysis with Student’s *t*-test are presented in Table 6.

### 3.4. Type of Surgery

In order to verify the hypothesis regarding the type of surgery and the perceived self-esteem and life satisfaction in the patients, an analysis was performed with the non-parametric Mann–Whitney U test. The analysis was carried out among the six types of treatments most often chosen by patients. The procedures included in the analysis concerned a breast implant, breast reconstruction, breast reduction, abdominoplasty, eyelid correction and birthmark excision. The results of the frequency of each type of surgery among individual groups are presented in Figure 1.

Among all types of procedures, the only statistically significant difference was in patients with a breast implant. In the group of patients assessing well-being before and after the procedure, the analysis with the Mann–Whitney U rank test showed a statistically significant difference between the groups, U = 451.00, *p* < 0.05, whereas in the case of the evaluation on the Flourishing Scale between the group of patients before and after the procedure, the difference was U = 460.50, *p* < 0.05.

### 3.5. Age Groups

In order to verify the research problem, the correlation analysis was performed using r, the Pearson method. In the first step of the analysis, the variable age was recoded according to the median criterion Me = 48. It turned out that among the patients up to 48 years of age, the body assessment was not significantly correlated with any form of well-being. Among patients after the surgery, the body assessment was correlated with all forms of well-being. The strongest correlation was between the scale of positive and negative emotions and the moderate body assessment, where r = 0.40; detailed results are presented in Table 7.

In patients over 49 years of age, the body assessment before the procedure was correlated to a low degree with the Life Satisfaction Scale, where r = 0.24. In the case of postoperative patients, the body assessment was moderately positively correlated with the life satisfaction scale, where r = 0.50; detailed results are presented in Table 8.

#### Graphical Presentation of Procedure Choices Depending on the Patients’ Education

Patients with a vocational education most often underwent abdominoplasty, whereas a smaller percentage chose the implantation of a breast implant and removal of a birthmark. The results of the frequency among individual groups are presented in Figure 2.

Patients with secondary education most often underwent breast implantation and breast reduction. The results of the frequency of each type of surgery among individual groups are presented in Figure 3.

Patients with post-secondary education most often underwent the procedure of the implantation of a breast implant and the removal of a birthmark. A smaller percentage chose abdominoplasty and breast reconstruction. The results of the frequency of each type of surgery among individual groups are presented in Figure 4.

Patients with a higher education most often underwent the procedure of inserting a breast implant, whereas a smaller percentage chose the procedure of removing a birthmark. The results of the frequency of each type of surgery among individual groups are presented in Figure 5.

The analysis of the average self-esteem among the patients before and after the surgery showed differences in self-esteem depending on the type of surgery. Patients who underwent abdominoplasty, eyelid correction and birthmark excision after the procedure had a higher self-esteem. In the case of implantation of a breast implant, the self-esteem of the patients was high, both before and after the procedure. The results are presented in Figure 6.

The analysis of the average assessment of well-being among the patients before and after the procedure showed differences in the assessment of well-being depending on the type of surgery. In all groups of patients, regardless of the type of procedure performed, the feeling of well-being in life increased after the procedure. The greatest increase in well-being was noted in patients who underwent breast implantation and birthmark excision. The results are presented in Figure 7.

The next analysis was of the mean of positive functioning in patients before and after a given type of surgery. Patients who decided to undergo breast implantation, abdominoplasty and eyelid correction after the procedure rated their daily functioning higher. The results are presented in Figure 8.

The analysis of the average positive and negative emotions experienced by the patients showed that patients who decided to have abdominoplasty, eyelid correction and birthmark excision after the procedure experienced more positive than negative emotions. The greatest increase in positive emotions over negative emotions was recorded in patients who underwent abdominoplasty. The results are presented in Figure 9.

The analysis of means concerning the subjective self-esteem with regard to the body in patients before and after the procedure showed an increase in the assessment of their own body in patients who underwent abdominoplasty. In the remaining groups of patients, the assessment of their own body before the procedure was higher than after the procedure. The results are presented in Figure 10.

The analysis of the average level of satisfaction with life, the level of flourishing and the perception of positive and negative emotions in patients before and after the procedure showed that in patients after the procedure, there was an increase in all well-being scales. The results are presented in Figure 11.

Basic descriptive statistics for the three types of well-being, self-esteem and subjective self-esteem of patients are presented in Table 9.

## 4. Discussion

In this study, in order to eliminate dissonance, measurements were repeated in postoperative patients several months after surgery when distractors such as surgical stress, postoperative swelling or objective visual effects were of less importance in the assessed patients. Due to the nature of the work, other surgical patients were not included as a control group. The introduction of such a group of patients could disrupt the study, since non-aesthetic surgical operations are primarily intended to improve functionality within a specific part of the patient’s body. The aesthetic aspect should not be ignored, however, for the purposes of this study; only the group of aesthetic medicine patients was taken into account.

### 4.1. The Procedure

The study showed an increase in the subjective self-assessment of the body in patients after surgery. The confirmation of the first hypothesis may mean that plastic and aesthetic surgery procedures positively shape self-esteem regarding the appearance of one’s own body. The results of this study regarding body image are consistent with the previous studies [15].

### 4.2. Self-Esteem

The study showed a slight increase in self-esteem after the surgery. Patients who underwent procedures were characterized by an average level of self-esteem. The postoperative increase in self-esteem observed in the patients was small, but it can be concluded that it was the result of fulfilled expectations set before the procedure. It is worth noting that the increase in self-esteem seems to be due to the motivation of the patients as well as the type of surgery. However, the relatively low increase in the self-esteem of the respondents after the procedure was probably caused by a different motivation than the internal motivation of the patients. The second factor causing a low increase in self-esteem may be related to procedures of a functional nature, such as eyelid correction or mole removal. Although it is impossible to ignore the aesthetic purpose of these treatments, their primary function was to restore certain places on the patient’s body to a state of comfort. This may result in an increase in the overall assessment of one’s own body with a relatively low increase in self-esteem if the given body element was not very burdensome, but only disturbing.

### 4.3. Age

The study showed a higher self-esteem with regard to one’s own body among the younger group of patients, both before and after surgery. The disproportion also persists after the procedure, with the difference being that in older patients a slight increase in self-esteem is noticeable, which does not occur in younger patients. The conducted research shows that although younger patients evaluate their bodies higher, their evaluation does not increase after the procedure. However, there were no differences in self-esteem between the groups. The study showed that all procedures, except for breast implants, increased the self-esteem of patients, whereas the insertion of breast implants caused a slight decrease in it. The decrease in the self-esteem of patients after the implantation of breast implants is probably related to motivational factors. Patients driven by the pressure of their environment and following models that promote large inserts on social media may be dissatisfied with the end result, which contributes to a lower self-esteem. In the case of patients under pressure from their environment, a decrease in self-esteem may also result from financial problems.

### 4.4. Type of Surgery

The study did not show that certain types of treatments improve the life satisfaction or self-esteem of patients more, with the exception of breast implantation. The patients’ psychological parameters improved regardless of the procedure performed. This may mean that a properly performed procedure has a positive effect on well-being and self-esteem.

### 4.5. Age Groups

The patients were divided into two age groups. Among the patients up to 48 years of age, there was a correlation between the subjective assessment of the body and the three examined forms of well-being after the procedure. No significant relationships were found in patients from this group before the procedure. This shows a significant relationship between the perception of one’s own body and the quality and satisfaction with life. In the case of patients over 49 years of age, a relationship between satisfaction with life and self-esteem both before and after the procedure was demonstrated; thus, special attention should be paid to improving the quality of life of patients after the procedure. A new factor that deserves further investigation is the breakdown of patients by age. The hypothesis concerning the relationship between the assessment of one’s own body and forms of well-being was confirmed in the part concerning patients up to 48 years of age. This indicates that the decision to undergo the procedure in this group of patients results in a better mood manifested by the prevalence of positive affects over negative ones. The increase in the Flourishing Scale also proves that patients have greater internal psychological resources, which contributes to reducing the negative emotional states of patients that lead to emotional disorders. In patients with a higher self-esteem with regard to their own body, positive emotions also outweigh negative emotions. In the case of patients over 49 years of age, the only demonstrated correlation between the assessment of one’s own body was with life satisfaction. This relationship occurred both before and after surgery. It is worth emphasizing the stronger correlation after the procedure. This seems to prove that patients play more social roles as they age in which they can find fulfillment. They have other priorities, where good looks complement the whole. These patients accept their bodies to a greater extent; however, similarly to younger patients, the procedure has a positive effect on their well-being [21,28].

The study focused on the emotional aspects of patients who underwent the procedure; however, the issue of financial resources and the cost of treatments in relation to these resources should also be considered during the study. The significant burden of the procedure cost could reduce or negatively affect the self-esteem and body assessment of the patient after the procedure, but bearing in mind that self-esteem is a relatively constant feature, it can be concluded that the question of finances would be useful but would not significantly affect the results of the study. In this work, standardized tools for measuring psychological variables were used. Due to the varied nature of the procedures performed, the statistical analysis did not use selective scales measuring satisfaction with specific parts of the body in comparison with other psychological factors. This limitation resulted from an insufficiently precise description by the patients of the procedure performed, and an indication by the person conducting the procedure would violate the anonymity of the study. In future works, it is worth expanding the research field to include factors motivating patients to undergo the procedure. This would be helpful in understanding the differences in patients’ lower self-esteem after surgery. Such a function would be fulfilled by an open question where the respondents could refer in a descriptive manner to their own motivation which prompted them to undergo the procedure. The research results presented above indicate that the supplement to the preoperative procedure algorithm should be a psychological consultation aimed at verifying the motivating factors in patients.

## 5. Results

The study showed that patients between 44 and 59 years of age most often used aesthetic and reconstructive medicine treatments.

The analysis of the number of patients undergoing aesthetic and reconstructive medicine procedures showed that the highest percentage is in the group with a middle and high education.

## 6. Conclusions

The conducted research shows that plastic surgery, apart from its aesthetic values, has a positive effect on the health of patients in the form of increasing well-being in its various forms and increased self-esteem.Plastic surgery can be helpful for patients with lowered psychological parameters resulting from dissatisfaction with their body image.Patients undergoing plastic surgery are most often satisfied with the changes in a specific part of their body.Psychological consultation following a surgical consultation would be helpful in determining the patients’ needs. This facilitates subsequent consultations with the surgeon and excludes the possibility of performing the procedure for unstable patients, which could adversely affect their subsequent assessment of the procedure.It has been shown that various forms of well-being correlate positively with the self-esteem of the female body. This means that the better the patient assesses a specific part of the body after the procedure, the higher the level of well-being in her life.

## Figures and Tables

**Figure 1 ijerph-19-11238-f001:**
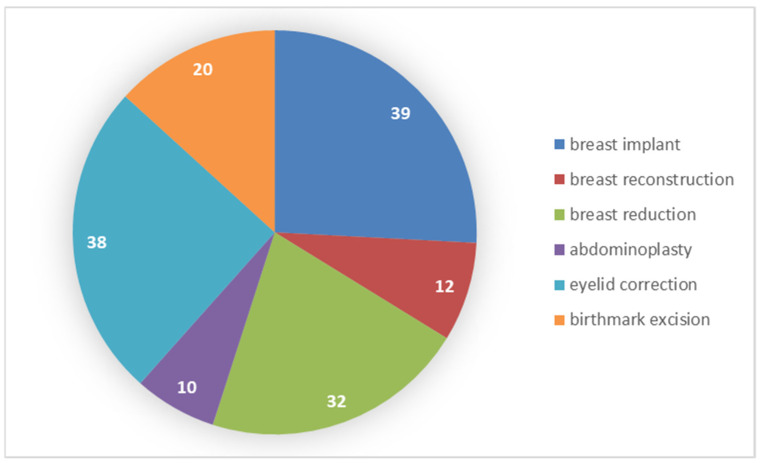
Number of patients in relation to the chosen type of surgery.

**Figure 2 ijerph-19-11238-f002:**
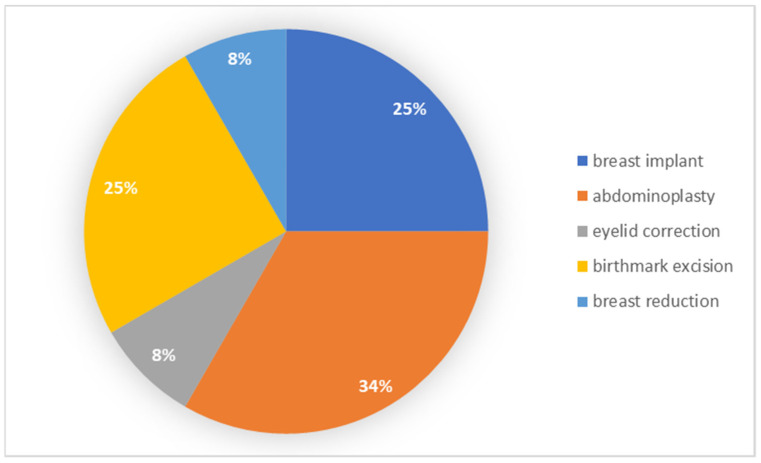
Types of procedures performed by patients with vocational education.

**Figure 3 ijerph-19-11238-f003:**
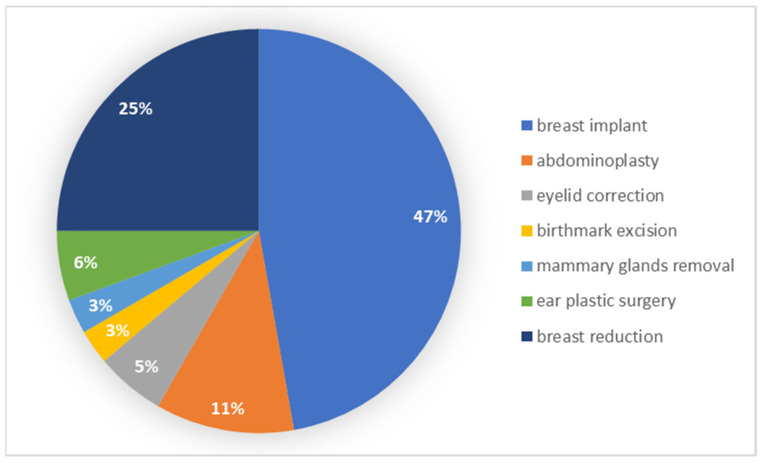
Types of procedures performed by patients with secondary education.

**Figure 4 ijerph-19-11238-f004:**
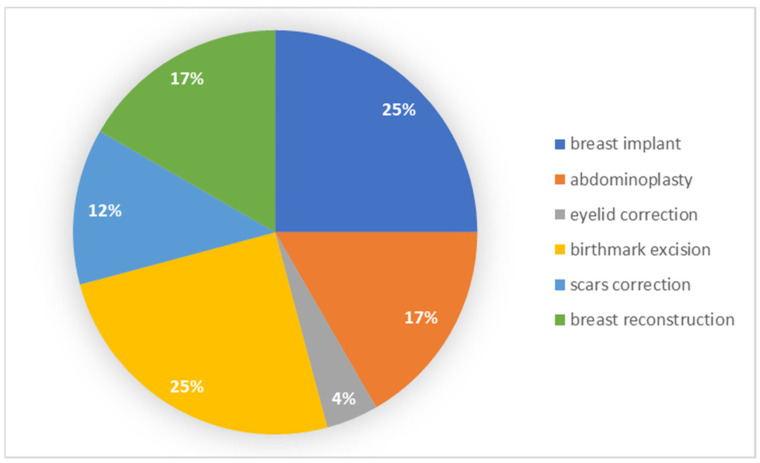
Types of procedures performed by patients with post-secondary education.

**Figure 5 ijerph-19-11238-f005:**
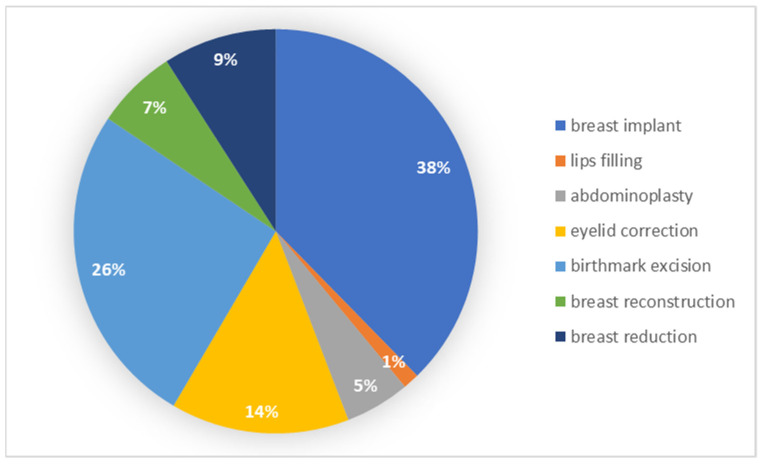
Types of procedures performed by patients with higher education.

**Figure 6 ijerph-19-11238-f006:**
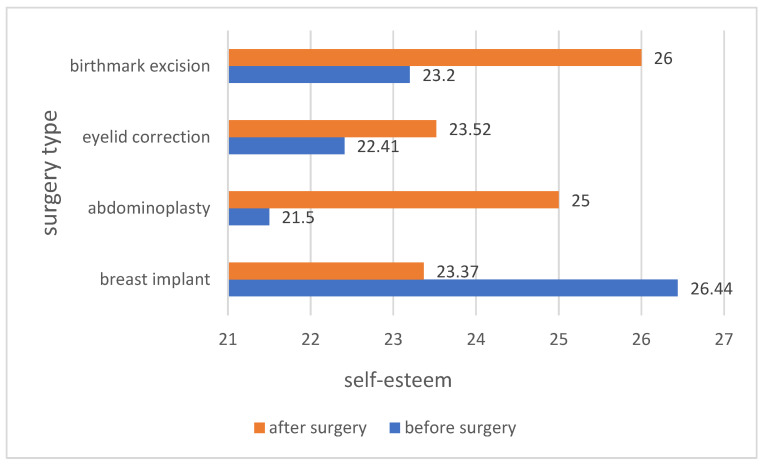
Comparison of patients’ self-esteem before and after a specific type of surgery.

**Figure 7 ijerph-19-11238-f007:**
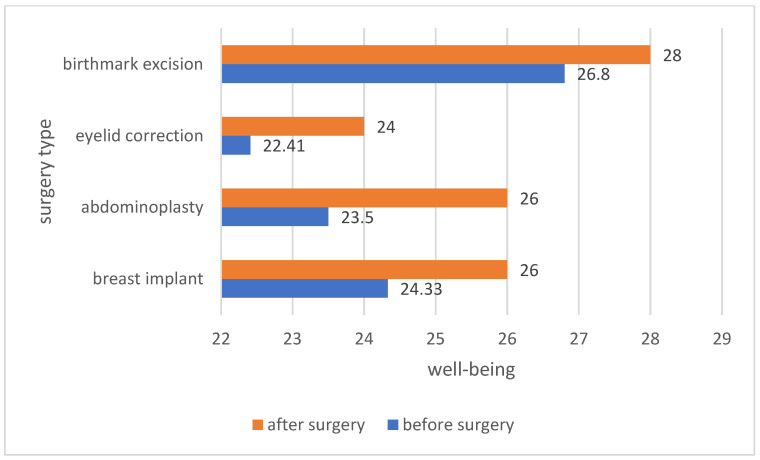
Comparison of the sense of well-being among patients before and after the procedure.

**Figure 8 ijerph-19-11238-f008:**
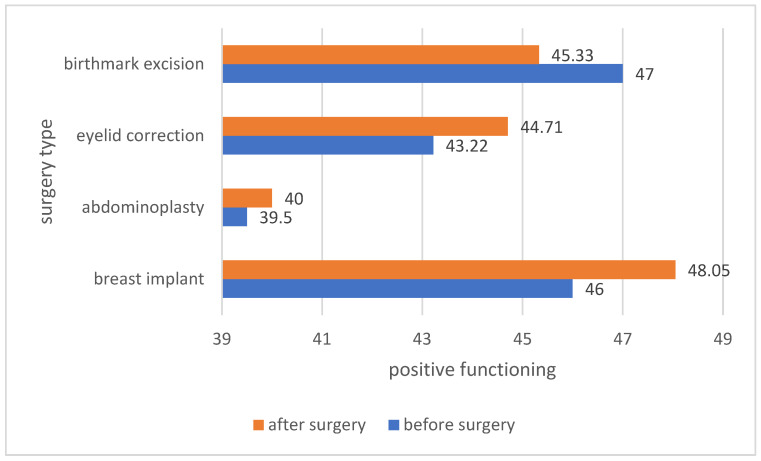
Comparison of the sense of positive functioning among patients before and after the procedure.

**Figure 9 ijerph-19-11238-f009:**
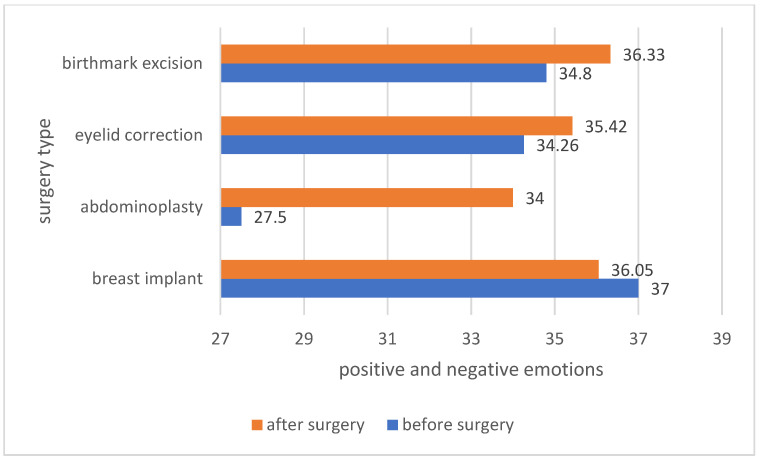
Positive and negative emotions felt among patients before and after the procedure.

**Figure 10 ijerph-19-11238-f010:**
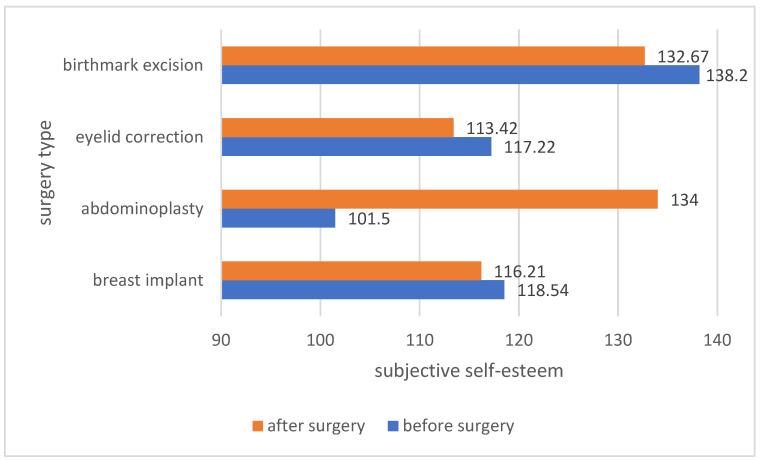
Comparison of the subjective self-esteem with regard to the patient’s body before and after the procedure.

**Figure 11 ijerph-19-11238-f011:**
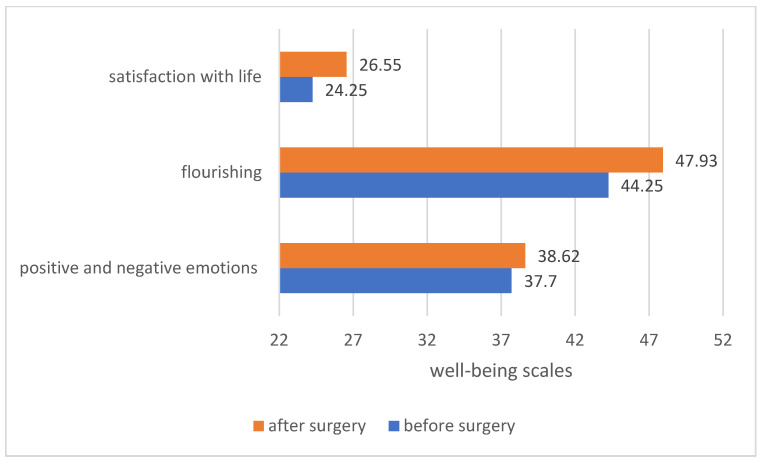
Comparison of various forms of patients’ well-being before and after the procedure.

**Table 1 ijerph-19-11238-t001:** Descriptive statistics of the subjective assessment of the patient’s own body before and after the procedure.

	Procedure	N	M	SD
Self-esteem	Before	116	176.88	22.64
	After	116	186.93	21.04

**Table 2 ijerph-19-11238-t002:** The results of the analysis with the use of the Student *t*-test of patients before and after the procedure for the variable: own body assessment.

	*F*	*t*	*df*	*p*
Self-esteem	1.92	−5.25	115	0.001

**Table 3 ijerph-19-11238-t003:** Descriptive statistics of patients’ self-esteem before and after the procedure.

	Procedure	N	M	SD
Self-esteem	Before	114	22.96	2.31
	After	114	23.61	1.98

**Table 4 ijerph-19-11238-t004:** Analysis of patients’ self-esteem using the Student *t*-test before and after the procedure.

	*F*	*t*	*df*	*p*
Self-esteem	0.24	−2.73	113	0.007

**Table 5 ijerph-19-11238-t005:** Descriptive statistics of the patient’s body assessment, divided by age, before and after the procedure.

Procedure	Age	N	M	SD
Before	Younger	85	130.67	1.99
	Older	79	115.37	2.96
After	Younger	64	129.78	2.21
	Older	62	117.35	3.14

**Table 6 ijerph-19-11238-t006:** Student *t*-test results for dependent samples in the group of women before and after the procedure and the level of assessment of their body.

Self-Esteem	*F*	*t*	*df*	*p*
Before	3.79	4.34	162	0.53
After	4.01	3.14	124	0.47

**Table 7 ijerph-19-11238-t007:** Correlations of various forms of well-being and body self-esteem in patients up to 48 years of age.

Women up to 48 Years of Age					
Procedure		SWLS	Flourishing Scale	SPANE	Body Evaluation
Before	SWLS	1	0.66 **	0.05	0.15
	Flourishing Scale	0.66 **	1	0.05	0.15
	SPANE	0.04	0.05	1	0.04
	Body evaluation	0.15	0.15	0.04	1
After	SWLS	1	0.53 **	0.40 **	0.35 **
	Flourishing Scale	0.53 **	1	0.31 *	0.31 *
	SPANE	0.40 **	0.31 *	1	0.40 **
	Body evaluation	0.35 **	0.31 *	0.40 **	1

** Correlation significant at the level of 0.01 (two-sided). * Correlation significant at the level of 0.05 (one-sided).

**Table 8 ijerph-19-11238-t008:** Correlations of various forms of well-being and body self-esteem in patients over 49 years of age.

Women over 49 Years of Age					
Procedure		SWLS	Flourishing Scale	SPANE	Body Evaluation
Before	SWLS	1	0.72 **	0.50 **	0.24 *
	Flourishing Scale	0.72 **	1	0.61 **	0.10
	SPANE	0.50 **	0.61 **	1	0.13
	Body evaluation	0.24 *	0.10	0.13	1
After	SWLS	1	0.62 **	−0.06	0.50 **
	Flourishing Scale	0.62 **	1	0.17	0.24
	SPANE	−0.62	0.17	1	0.14
	Body evaluation	0.50 **	0.24	0.14	1

** Correlation significant at the level of 0.01 (two-sided). * Correlation significant at the level of 0.05 (one-sided).

**Table 9 ijerph-19-11238-t009:** Basic descriptive statistics of the researched quantitative variables.

		Breast Implant	Abdominoplasty	Eyelid Correction	Birthmark Excision
Self-esteem					
before surgery	N	39	11	39	20
	M	26.44	21.50	22.41	23.20
	SD	4.25	1.44	2.52	2.07
after surgery	N	51	9	34	22
	M	23.37	25.00	23.52	26.00
	SD	1.89	1.83	1.81	2.31
Wellbeing					
before surgery	N	39	11	39	20
	M	24.33	23.50	22.41	26.80
	SD	5.42	3.68	5.27	4.88
after surgery	N	51	9	34	22
	M	26.00	26.00	24.00	28.00
	SD	5.28	5.50	4.77	3.44
Sense of positive functioning					
before surgery	N	39	11	39	20
	M	46.00	39.50	43.22	47.00
	SD	6.59	5.25	8.48	6.47
after surgery	N	51	9	34	22
	M	48.05	40.00	44.71	45.33
	SD	5.71	4.39	4.78	6.97
Positive and negative emotions					
before surgery	N	39	11	39	20
	M	37.00	27.50	34.26	34.80
	SD	3.66	5.52	4.87	3.31
after surgery	N	51	9	34	22
	M	36.05	34.00	35.42	36.33
	SD	4.51	3.41	5.29	3.38
Subjective self-esteem					
before surgery	N	39	11	39	20
	M	118.56	101.50	117.22	138.20
	SD	20.57	14.94	27.75	14.73
after surgery	N	51	9	34	22
	M	116.21	134.00	113.42	132.67
	SD	21.14	13.14	29.50	17.49

## Data Availability

Not applicable.

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
