# Peer review of "Welfare and Self-Assessment in Patients after Aesthetic and Reconstructive Treatments"

_ijerph, 2022, doi:10.3390/ijerph191811238_

Round 1
Reviewer 1 Report
The paper examines an interesting topic, one that relates to modern society given the elevated beauty standards we see in the media. However, the paper requires major revision before it is deemed acceptable for publication in IJERPH.
In the introduction on page 2, the authors mention that previous research shows that after plastic surgery, people are more satisfied with their own appearance and experience an increase in quality of life than before surgery. This poses an important question: What is the main contribution of this study? In my opinion, there seems to be an interesting angle in research questions 3, 4, and 5 – the authors may consider positioning the paper along those lines (i.e., age, type of surgery and age groups).
It is also important to point out that what the authors refer to as hypotheses are actually research questions as the authors do not really offer any prediction. Hypotheses are generally developed using previous literature and are statements with clear predictions.
The paper lacks an important part of any academic paper – a theoretical background and hypothesis development sections.
The Methodology section needs refinement in the way it is presented. The sample size is actually 126 respondents and not 290 as only 126 women completed the repeated measure questionnaire.
As for the results, the authors could benefit from using a repeated measure on SPSS to better understand the correlation between surgery and the dependent variables. Repeated measures offer a more accurate analysis of longitudinal studies. It is very important to explain why when measuring subjective assessment and self-esteem the sample size for the analyses were only 116 and 114 respectively out of the 126. The authors could also benefit from presenting a separate analysis for positive and negative emotions rather than combine them. These emotions generally go in opposite directions when measured simultaneously so the results will vary.
Up until the discussion the paper reads like a Nielsen report rather than an academic paper. The discussion section taps into the theoretical background on aesthetic surgeries and that is good, but should not be placed in the discussion section, but rather in the theoretical background and hypothesis development sections. The discussion should be more of an overview of the paper and its overarching theme based on the results obtained.
On a minor note, the authors could benefit from proofreading the paper as it has some grammatical mistakes.
Author Response
In the introduction on page 2, the authors mention that previous research shows that after plastic surgery, people are more satisfied with their own appearance and experience an increase in quality of life than before surgery. This poses an important question: What is the main contribution of this study? In my opinion, there seems to be an interesting angle in research questions 3, 4, and 5 – the authors may consider positioning the paper along those lines (i.e., age, type of surgery and age groups).
The aim of the work was to learn about the psychological parameters of patients undergoing aesthetic surgery and to answer the question whether it would be advisable for the conducting algorithm to introduce obligatory psychological consultations for patients who wish to undergo aesthetic procedures.
It is also important to point out that what the authors refer to as hypotheses are actually research questions as the authors do not really offer any prediction. Hypotheses are generally developed using previous literature and are statements with clear predictions.
Thank you for this comment, it has been corrected.
In the introduction on page 2, the authors mention that previous research shows that after plastic surgery, people are more satisfied with their own appearance and experience an increase in quality of life than before surgery. This poses an important question: What is the main contribution of this study? In my opinion, there seems to be an interesting angle in research questions 3, 4, and 5 – the authors may consider positioning the paper along those lines (i.e., age, type of surgery and age groups).
Thank you very much for your attention. Our goal in the first part was to show the self-esteem of the body and the self-esteem of the patients before and after the surgery, hence the remaining indicated points took further places.
It is also important to point out that what the authors refer to as hypotheses are actually research questions as the authors do not really offer any prediction. Hypotheses are generally developed using previous literature and are statements with clear predictions.
Thank you for your attention, the hypotheses have been modified as suggested.
The paper lacks an important part of any academic paper – a theoretical background and hypothesis development sections.
Thank you for your comments, the theoretical basis has been included in the introduction.
The Methodology section needs refinement in the way it is presented. The sample size is actually 126 respondents and not 290 as only 126 women completed the repeated measure questionnaire.
Thank you very much for your comments, the methodology section has been improved according to your suggestions.
As for the results, the authors could benefit from using a repeated measure on SPSS to better understand the correlation between surgery and the dependent variables. Repeated measures offer a more accurate analysis of longitudinal studies. It is very important to explain why when measuring subjective assessment and self-esteem the sample size for the analyses were only 116 and 114 respectively out of the 126. The authors could also benefit from presenting a separate analysis for positive and negative emotions rather than combine them. These emotions generally go in opposite directions when measured simultaneously so the results will vary.
Thank you very much for your comments. When measuring subjective body self-esteem and self-esteem, the sample sizes of 116 and 114, respectively, resulted from too many questions left unanswered in the questionnaire. The use of combined analysis for positive and negative emotions was dictated by the standardization of the questionnaire, which assumed checking both poles of emotions together.
Up until the discussion the paper reads like a Nielsen report rather than an academic paper. The discussion section taps into the theoretical background on aesthetic surgeries and that is good, but should not be placed in the discussion section, but rather in the theoretical background and hypothesis development sections. The discussion should be more of an overview of the paper and its overarching theme based on the results obtained.
Thank you very much for your comments. As suggested, the theoretical basis has been moved from the discussion section to the introduction section.
Reviewer 2 Report
The styles of the referencing in the text and in the bibliography do not match. The order of the in-text references is a total mess. This makes the review nearly impossible to proceed the review process.
Author Response
Thank you very much for your careful attention.
All the shortcomings in the bibliography have been corrected.
Round 2
Reviewer 2 Report
Significant improvements are recognized in the revised manuscript. I hope this article may help the readers further understand of issues dealt with in the manuscript.